# Improving Water Efficiency in a Municipal Indoor Swimming-Pool Complex: A Case Study

**Flora Silva** [1,5,6,*], **Ana M. Antão-Geraldes** [2,5,*], **Carmem Zavattieri** [1,3], **Maria João Afonso** [1], **Flávio Freire** [3] and **António Albuquerque** [4,5,6]

1   ESTiG, Instituto Politécnico de Bragança, Campus de Santa Apolónia, 5300-253 Bragança, Portugal; carmemzavattieri@hotmail.com (C.Z.); mjafonso@ipb.pt (M.J.A.)
2   Centro de Investigação de Montanha (CIMO), Instituto Politécnico de Bragança, Campus de Santa Apolónia, 5300-253 Bragança, Portugal
3   Universidade Tecnológica Federal do Paraná, Campus Curitiba, Curitiba 80230-000, PR, Brazil; freireutfpr@gmail.com
4   Departamento de Engenharia Civil e Arquitectura, Universidade da Beira Interior, 6201-001 Covilhã, Portugal; antonio.albuquerque@ubi.pt
5   FibEnTech, 6201-001 Covilhã, Portugal
6   GeoBioTec-UBI, 6201-001 Covilhã, Portugal
*   Correspondence: flora@ipb.pt (F.S.); geraldes@ipb.pt (A.M.A.-G.)

**Abstract:** This study aimed to determine the water demand of a municipal swimming pool complex to propose water use efficiency measures. Concomitantly, the possibility of recycling and reusing the water from filter backwashing was evaluated. The pools consumed 25.6% of water, the filter backwashing 24.5%, and the showers 34.7%. Despite the current impossibility of reducing water consumption in pools and filter backwashing, it is feasible to promote more efficient use of water through reducing water consumption by adopting simple water-saving initiatives for showers, taps, and flushing cisterns. These were organized into three distinct scenarios: (a) flushing cistern volume adjustment and the replacement of washbasin and kitchen taps; (b) flushing cistern volume adjustment and shower replacement and (c) flushing cistern volume adjustment, shower, washbasin, and kitchen taps replacement. Under scenarios 1, 2, and 3, the water consumption reduction was 8.0, 13.2, and 20.4%, respectively. The initial investment for scenario 1 was €2290.5, €859.0 for scenario 2 and €3149.5 for scenario 3; the annual water bill reduction was €7115.4, €11,518.1, and €17,655.9, respectively. Therefore, the turnover of the investment was four (scenario 1), one (scenario 2), and three months (scenario 3). The filter washings attained the required standard for irrigation after being subjected to 15 h of sedimentation.

**Keywords:** water efficiency; indoor municipal swimming pools; economic viability; filter washings reuse





## 1. Introduction

It is generally known that the ecological services provided by aquatic ecosystems and the water industry are facing increasingly complex challenges: the demand for water is continuously rising, causing the rapid exhaustion of existing water resources. Urban population growth and changing lifestyles, implying more intensive water use [1] and increasing water pollution [2], are the leading causes of this phenomenon. Besides, future climate changes will involve higher temperatures and changes in the intensity and patterns of precipitation, leading to more frequent droughts, as well as reducing water quantity and quality, in southern Europe and the Iberian Peninsula [3,4]. Thus, the urban water cycle is affected by climate change, but it is also contributing to climate change. Indeed, water potabilization processes, the delivery of water to consumers, and the treatment of wastewater use significant amounts of energy, contributing to increasing $CO_2$ and other greenhouse gas emissions [1,5,6]. Therefore, in the face of this scenario, it is important

to implement water sustainability, promoting water use efficiency by reducing water consumption through the adoption of efficient products or devices (e.g., taps and showers), reducing waste and losses, and by reusing and recycling water [7]. Public indoor swimming-pool facilities use large amounts of water and, consequently, energy, because of their particular characteristics: (1) the relatively high temperature and humidity levels in the pool room; (2) the evaporation caused by pool usage; (3) the use of warm water for pools and showers; and (4) the requirement of a water treatment system [8,9]. Furthermore, users' number and behavior, the variety of services provided, the operating patterns [8,9], and the quality of indoor water and air [10,11] also influence the aforementioned characteristics. Nevertheless, to our best knowledge, research concerning the impact of these facilities on urban water demand is scarce. Some research has focused on residential swimming pools, revealing that pools have a very high impact on urban water demand and consequently on ecological integrity and the services provided by freshwater ecosystems [12–15]. However, approaches concerning municipal and public indoor swimming pools are even scarcer and, in some cases, very preliminary [16–19]. Similarly, research concerning the recycling and reuse feasibility of pool filter backwashing is also very scarce [20–24]. Therefore, to contribute to filling this information gap, the aim of this research is: (a) to determinate the water demand of a municipal indoor swimming pool complex; (b) to propose water use efficiency measures, analyzing their feasibility and costs; (c) to evaluate the possibility of recycling and reusing the water from swimming-pool filter backwashing.

Municipalities may use the results of this study to implement and improve measures to achieve greater water use efficiency in their infrastructure.

## 2. Materials and Methods

### 2.1. The Case Study

Bragança (Latitude: 41°48′20″ N; Longitude: 6°45′25″ altitude: 673 m) is the principal city of northeast Portugal, with 24,078 inhabitants [25]. The climate is continental with Mediterranean influences. Annual mean precipitation is around 700 mm per year, occurring mainly in autumn and winter, but in a very irregular pattern [26]. Two reservoirs designed with a 99% supply guarantee rate are the base of the urban water supply. However, because this value was calculated without considering the changing climate conditions, it will likely be insufficient for future water supply [27]. According to the information provided by the municipality, the water demand per year is around 3757.615 m$^3$. Approximately 80% of this water is for domestic consumption, whereas the remainder is for commercial and industrial purposes. Indeed, Bragança is one of the Portuguese cities with the highest water demand: more than 262 L/(inhabitant/day) [28].

The Bragança indoor municipal swimming pool complex was launched in 2003. This complex includes two covered pools (competition and learning pools), with barriers around the pool to collect and reuse any splashes or overflows (Figure 1; Table 1) and the following installations: male, female, and staff dressing rooms, showers, WCs, Café, first aid room.

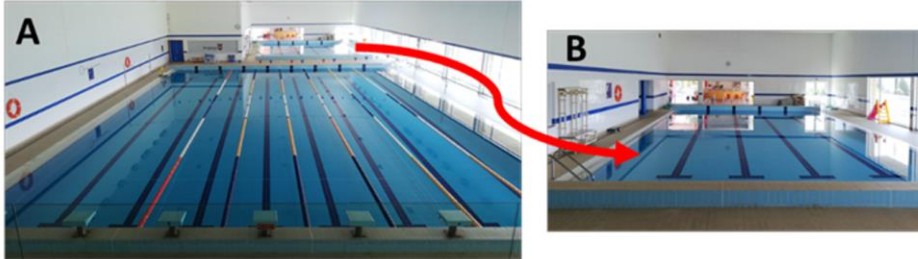

**Figure 1.** General view of the competition (**A**) and learning (**B**) pools.

**Table 1.** Swimming-pool's characteristics and mean values of some water parameters (Source: Bragança Municipality).

| Characteristics | Competition | Learning |
|---|---|---|
| Air temperature °C | 30 | 30 |
| Air humidity (%) | 50–60 | 50–60 |
| Dimensions (L × W) (m) | 25 × 17 | 16.6 × 10 |
| Base area ($m^2$) | 425.0 | 166.6 |
| Minimum depth (m) | 1.8 | 0.7 |
| Maximum depth (m) | 2.0 | 1.3 |
| Water volume ($m^3$) | 890.0 | 160.0 |
| Water added ($m^3$/day) | 0.7 | 0.3 |
| Filter type | Pressure, closed | Pressure, closed |
| Number of filters | 2 | 2 |
| Filter diameter (mm) | 2000 | 1400 |
| Type of filter bed | sand | sand |
| Filtration surface ($m^2/ft^2$) | 3.1/33.8 | 1.5/16.6 |
| Filtration speed (L/min/$m^2$) | 500 | 500 |
| Flow rate (L/min) | 1566.7 | 766.7 |
| Water residence time (h) | 4.2 | 2.1 |
| Water temperature (°C) | 26–28 | 26–28 |
| Water conductivity (µS/cm) | 190 | 200 |
| pH | 7.8 | 7.7 |
| Water turbidity (NTU) | <0.5 (l.q.) | <0.5 (l.q.) |
| Total chlorine (mg $Cl_2$/L) | 1.1 | 1 |
| Free chlorine (mg $Cl_2$/L) | 0.9 | 0.8 |
| Combined chlorine (mg $Cl_2$/L) | 0.2 | 0.2 |
| Trihalomethanes * (µg/L) | <7 (l.q.) | <7 (l.q.) |

l.q.: Limit of quantification * Analytical methodology: PAFQ17 (Portuguese Accreditation Institute-IPAC).

The water for the pools comes from the public water supply system. It is simultaneously heated and treated with chlorine through the recirculation system, and continuously recycled in sand filters (Table 1). These filters are backwashed every two days. According to information provided by the Municipality concerning the period from 2016–2018, the average number of users per year is 45,957. October and November are the months with the highest number of users, with an average of 6142 users per month, whereas August and September are the months with the fewest users (732 users per month). The closure of the complex between these two months for pool maintenance and cleaning explains the latter value.

*2.2. Water Efficiency Determination*

The water flow rates in the taps and showers were obtained by recording the duration of filling a specific volume "X" in a 10 L capacity bucket. Next, with the aid of a calibrated beaker, the final volume of water in the bucket was determined. The volumes of flushing cisterns were measured by registering the maximum level, producing a discharge, and then filling the cistern manually with a calibrated container until the restoration of the initial level. To obtain the water flow rates in the urinals, their outlets were covered with adhesive tape. Simultaneously, when the flow meter was activated, the discharge duration was measured in seconds. Finally, the water in the outlet was measured using a calibrated beaker. Concomitantly, to determine the water use behavior and patterns, a survey (Appendix A) was addressed to 70 swimming pool users to calculate the total average flow of each water device. Considering the obtained results for the water consumption, the flow rates of these water devices were adjusted to the reference values of class "A" water efficiency (the highest efficiency) of the Portuguese labeling system [29]. Nevertheless, when used in public buildings, their application is not very strict for health reasons, to ensure the proper functioning of drainage systems and adequate levels of user comfort [7]. Therefore, when possible, suggestions for more efficient devices were proposed to achieve higher water

efficiency. The proposals considered the Portuguese labeling system recommendations concerning public buildings (efficiency, price, and ease of installation).

### 2.3. Water Use Efficiency Measures

The proposed measures were organized in scenarios (Table 2) and subsequently analyzed using the obtained data concerning water use (Section 2.2). The cisterns' flush volume regulation was included in all the scenarios, since this measure does not imply any additional costs. Concomitantly, the economic viability of each of the listed measures was analyzed. This approach considered the initial investment to implement the proposed water use efficiency measures, the reduction of water bills (due to the implementation of water use efficiency measures), and the turnover period.

**Table 2.** Proposed scenarios for reducing water expenditure in the Bragança municipal swimming pool complex.

| Scenarios | Measures |
|---|---|
| 1 | — Flushing cistern volume adjustment<br>— Replacement of washbasin and kitchen taps |
| 2 | — Flushing cistern volume adjustment<br>— Replacement of showers |
| 3 | — Flushing toilet volume adjustment<br>— Replacement of washbasin and kitchen taps<br>— Replacement of showers |

### 2.4. Reuse of Filter Backwashing

The amount of water necessary for backwashing the swimming pool filters was quantified by registering the values of the water flow meter immediately before and after the rinsing process. Four values were taken to obtain the mean total water consumption. The filters were rinsed by every two days for five minutes. To evaluate the possibility of reusing backwashed water for irrigating the lawn of the local football stadium, located in the surroundings of the swimming pool complex (Figure 2), samples from the filter backwashing were collected in the first and the third minute after the cleaning process started.

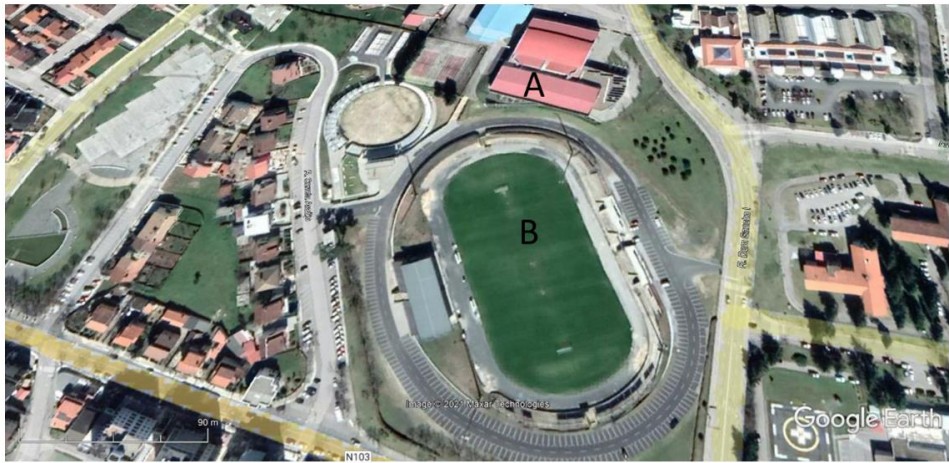

**Figure 2.** Plan of the Bragança swimming pool complex (**A**) and football stadium (**B**) (source: Google Earth).

In the laboratory, two sets of backwashed water were submitted to 2 h of sedimentation, two sets were submitted to 15 h of sedimentation, and the last two sets of samples were not submitted to sedimentation. Afterward, the following parameters were deter-

mined in triplicate according to [30]. The pH was measured (Method 4500 – H+) with a digital potentiometer (Inolab pH meter, WTW, Weilheim, Germany). The total suspended solids (TSS) were determined by the gravimetric method (Method 2540 D) using an oven (Series 9000, Scientific, Sandton, Gauteng, South Africa). The turbidity was determined by the nephelometric method (Method 2130 B) with a turbidimeter (Turb 550, WTW, Weilheim, Germany). The concentration of free residual chlorine was determined spectrophotometrically (spectrophotometer V-530, Jasco, Hachioji, Japan) by using the colorimetric method with N-Diethyl-p-phenylenediamine (Method 4500-Cl G). Furthermore, the enumeration of the total coliforms, fecal coliforms, and streptococci colonies was carried out by using the membrane filtration method according to ISO 9308-1:2000, an internal method, and ISO 7899-2:2000, respectively. The obtained results were compared to those in [31–33].

## 3. Results

### 3.1. Water Use Characterization

The estimated current average annual water consumption in the pool's building was 16,560.4 m$^3$, as shown in Table 3.

**Table 3.** Average annual water consumption in swimming pools by use category.

| Equipment/Activity | Consumption (m$^3$/year) | Percentage (%) |
| --- | --- | --- |
| Flushing cisterns | 413.6 | 2.50 |
| Showers | 5751.5 | 34.72 |
| Taps | 1950.2 | 11.78 |
| Urinals | 4.3 | 0.03 |
| Pools | 4243.2 | 25.62 |
| Filters backwashing | 4197.6 | 25.35 |
| Total | 16,560.4 | 100 |

Regarding the main findings of the water consumption calculations for sanitary and hygienic purposes (based on the monitoring of the water use and on the survey on how and when people use the water devices), the proposed audits were as follows.

3.1.1. Flushing Cisterns Complete Discharge (N = 28)

The flushing cisterns were all of the same model and brand. The average discharge volume was 9 L. Therefore, according to the survey results, there was one discharge/user/visit and the average number of users per year was estimated at 45,957. Thus, the estimation of the flushing cisterns' total annual water consumption was 413.6 m$^3$. Nevertheless, by simply regulating the inlet water mechanism, it is possible to reduce the current discharge to 6 L. Therefore, the average annual water consumption would decrease to 275.7 m$^3$, implying a reduction of around 33%.

3.1.2. Showers

- With timer (N = 25): the current average flow rate was 9.96 L/min. However, it can be reduced to 5.7 L/min with more efficient showerheads. The cost of these showerheads is 15.39 € + VAT per unit (according to 2020 prices);
- Mixer (N = 6) and single-lever (N = 2): the current flow rate was 23.13 and 19.60 L/min, respectively. Nevertheless, the flow rates could be reduced to values between 5 and 7.2 L/min if more efficient models were substituted for the current showerheads. The cost of the more efficient models is 22.70 € + VAT per unit (according to 2020 prices).

Considering that each user takes two to five minutes to shower (according to the survey data), the annual number of users, the average flow of each shower model, and the frequency of utilization of each model, the current average water consumption was

estimated at 5751.5 m$^3$. Therefore, a reduction to 3704.8 m$^3$ (35.59%) would be expected with the showerheads' replacement.

### 3.1.3. Taps

- Single lever tap (N = 15): these taps had an average flow of 17.38 L/min;
- Mixer (N = 10): these taps had an average flow of 3.33 L/min, which is considered highly efficient. However, this value could be misleading because we observed that users experienced difficulties in turning off the taps; often, after their use, they remained turned on, inducing large amounts of water wastage;
- Lever tap (N = 5): the average flow was 20.40 L/min;
- Medical tap (N = 3): the average flow was 8.90 L/min;
- Kitchen tap (N = 1): the average flow was 15.00 L/min.

According to the survey, each user used the tap once for 3 minutes, and the kitchen tap was used six times per day for 5 minutes; considering the annual number of users, the average flow of each tap model, and the frequency of utilization, the current average water consumption is 1950.2 m$^3$. Therefore, a reduction to 758.4 m$^3$ (61.1%) is expected with tap replacement. The taps could be replaced by more efficient and similar taps with an average flow of 5 L/min, and of 8 L/min in the case of the kitchen tap. Since these taps supply cold and hot water, the application of calibrated reducers was not considered because the opening time (between 6 and 8 s) was not long enough for hot water to arrive at the tap, leading to successive actuations and, consequently, wasting of water. As a precaution, it is recommended that the medical taps, even those labeled in the inefficient category, are kept because the specifications cover only washbasins and kitchen taps. Furthermore, it is expected that their use will be low [29].

### 3.1.4. Urinals

A 0.9 L flush volume urinal is in the water efficiency category "A ++" [29]. Therefore, these devices do not need to be adjusted or replaced.

### 3.2. Proposed Measures and Their Viability to Promote Water Use Efficiency

As mentioned in Table 3, these measures were organized in scenarios. The estimated water consumption reduction after implementing the water-saving measures proposed for each is presented in Table 4.

**Table 4.** Water consumption without and with water-saving measures.

| Scenarios | Consumption without Measures | Predicted Consumption with Measures | Predicted Reduction | |
|---|---|---|---|---|
| | (m$^3$) | (m$^3$) | (m$^3$) | (%) |
| 1 | 2363.9 | 1034.2 | 1329.7 | 8.03 |
| 2 | 6165.1 | 3980.6 | 2184.6 | 13.19 |
| 3 | 8115.4 | 4739.0 | 3376.4 | 20.39 |

In order to reduce water consumption, scenarios 2 and 3 are the most effective, since both include shower replacement measures. Indeed, the showers were the devices that presented the highest water consumption. The predicted costs for the implementation of each scenario are €2290.51 for scenario 1, €858.97 for scenario 2, and €3149.48 for scenario 3. It should be stressed that the regulation of flushing cisterns imposes no costs on the municipality. The annual water bill reduction would be €7115.44 (scenario 1), €11,518.11 (scenario 2), and €17,655.92 (scenario 3). The turnover of the investment for scenarios 1, 2, and 3 would be four, one, and three months, respectively. Indeed, the turnover periods obtained were considerably low—less than one year for all the scenarios. Despite presenting the largest initial investment, scenario 3 is the most effective, promoting higher water savings. The predicted turnover for scenario 3 can be justified by including measures from both scenarios 1 and 2. Indeed, the replacement of taps (scenario 1) combined with

shower replacement (scenario 2) represents the highest initial investment. Nevertheless, these costs can be offset by the reduction in water consumption. Therefore, one year after implementing the water-saving measures, the predicted turnover for scenarios 1, 2, and 3 is €4824.93, €10,659.14, and €14,506.44, respectively. Moreover, the predicted water consumption reduction can reach 20.39% (Table 4).

### 3.3. Preliminary Analysis of the Viability of Filter Backwashing Reuse

The average daily water consumption for swimming pool filter cleaning was 15.7 $m^3$ (competition pool) and 7.7 $m^3$ (learning pool). These filters (Table 1) are cleaned every two days. Therefore, the estimated annual water consumption for filter backwashing was 2818.8 $m^3$ and 1378.8 $m^3$, respectively, totaling 4197.6 $m^3$, which are discharged into the wastewater drainage network.

The highest values for the TSS, turbidity, and free residual chlorine concentrations were observed in the samples collected during the first minute of the filter backwashing (first flux). In those collected in the third minute, a reduction between 10% and 20% occurred in the studied parameters. Concerning the TSS, the 2 and the 15 hour sedimentation reduced the amounts of suspended solids to be 60% and 80%, respectively. A similar trend for both the turbidity and the free residual chlorine was recorded. Nevertheless, the TSS, turbidity, and free chlorine samples exceeded the admissible values, except those submitted to 15 hour sedimentation (Table 5). The pH values were always within the admissible range. The total coliforms, fecal coliforms, and fecal streptococci were not detected.

**Table 5.** TSS, turbidity, free chlorine, and pH values observed in 1st and 3rd minute. Filter-backwashed water samples were submitted to different sedimentation times.

| Parameter | Sample Sedimentation Time | Values [1,2,3] |
|---|---|---|
| TSS (mg/L) | No sedimentation (1 min) | 61.5 |
| | Sedimentation 2 h (1 min) | 24.5 |
| | Sedimentation 15 h (1 min) | 9.0 |
| | No sedimentation (3 min) | 49.5 |
| | Sedimentation 2 h (3 min) | 21.5 |
| | Sedimentation 15 h (3 min) | 10.0 |
| Turbidity (NTU) | No sedimentation (1 min) | 36.1 |
| | Sedimentation 2 h (1 min) | 5.0 |
| | Sedimentation 15 h (1 min) | 1.7 |
| | No sedimentation (3 min) | 25.5 |
| | Sedimentation 2 h (3 min) | 5.9 |
| | Sedimentation 15 h (3 min) | 1.9 |
| Free residual chlorine (mg/L) | No sedimentation (1 min) | 0.58 |
| | Sedimentation 2 h (1 min) | 0.12 |
| | Sedimentation 15 h (1 min) | 0.02 |
| | No sedimentation (3 min) | 0.42 |
| | Sedimentation 2 h (3 min) | 0.11 |
| | Sedimentation 15 h (3 min) | 0.02 |
| pH | No sedimentation (1 min) | 6.7 |
| | Sedimentation 2 h (1 min) | 7.1 |
| | Sedimentation 15 h (1 min) | 7.8 |
| | No sedimentation (3 min) | 6.8 |
| | Sedimentation 2 h (3 min) | 6.4 |
| | Sedimentation 15 h (3 min) | 7.8 |

[1] Flushing cistern quality requirement: TSS 10 mg/L; Turbidity 2 NTU [31]. [2] Irrigation quality requirement: TSS 10 mg/L; Turbidity 2 NTU [31]. Irrigation of private gardens: for lawn irrigation, the same requirements were adopted. [3] Recommended limit: Free residual chlorine 0.2–0.6 mg/L [32,33]; pH: 6.5–9.0 [32].

Considering a scenario including: (1) flushing cistern volume adjustment; (2) filter backwashing reuse to irrigate the lawn of the local football stadium; (3) the replacement

of showers; and (4) the replacement of the washbasin and kitchen taps, the water savings could reach 23.3%.

## 4. Discussion

The approach discussed in this study showed that it is possible to promote a substantial increase in water efficiency in this facility by simply increasing the efficiency of water use for sanitary and hygienic purposes. As in other swimming pool facilities, the showers demonstrated the highest water consumption [1,8,9,19,34]. Therefore, the proposed measures, if implemented, will reduce the financial burden and increase the environmental benefits associated with water use. Herein, the energy consumption under the scope of the water–energy nexus was not evaluated. However, several authors describe large amounts of energy consumption due to excessive water use in residential or public buildings [1,6,7,35–38]. According to data obtained by [7], each cubic meter of water consumed implies the consumption of 1.115 kWh in the water supply system and the consumption of 0.818 kWh in drainage treatment and wastewater treatment processes, adding up to a total of 1.933 kWh. Therefore, assuming similar levels of energy consumption in the Bragança region, it is plausible to admit that the water consumption reductions predicted in each scenario would lead to energy savings in the public system ranging from around 2562.96 to 6532.52 kWh/year. Consequently, the turnover and the earnings for the different studied scenarios are monetarily and environmentally much higher than the estimates obtained herein. Indeed, implementing inexpensive and straightforward water efficiency measures (substituting showers and taps with more efficient devices) is also an essential step to improving energy efficiency while simultaneously reducing the waste of water, $CO_2$, and other greenhouse gas emissions.

The results concerning water consumption in pools (Table 2) are underestimates because they do not include the water losses caused by evaporation, swimmers exiting the pool, and water splashes. The authors of [34] demonstrated that an indoor heated 300 m$^2$ pool can lose 21,000 L of water per week through evaporation (water temperature: 28 °C, air temperature: 29 °C, and air humidity: 60%). Indeed, the water consumption in an indoor swimming pool, excluding sanitary and hygienic requirements, can reach 160 L/person per day, and the energy used is reported to be between 400 and 1600 kWh/m$^2$ of the usable area [9,34]. In general, indoor swimming pools use two types of energy sources: (1) thermal energy for showers, pool water, and space heating (the most significant portion of energy consumption); and (2) electricity to power water pumping systems, lighting systems, rotating equipment, air cooling, dehumidifying processes, and water treatment systems [19]). In Bragança swimming pools, natural gas boilers are the equipment used for producing hot water. According to information provided by the municipality concerning the period from 2018–2019, the mean annual value of natural gas consumption in the swimming pool's facilities was 719 KWh/m$^2$ (the data concerning electricity consumption were not available). The most frequent water and energy saving measure is the application of a cover over the water's surface during periods in which the facility is not in use, allowing the reduction of water evaporation [19] and the release of volatile disinfection by-products that negatively affect indoor air quality, often causing eye irritation or even asthma in users [11,39]. Additionally, this measure reduces air humidity, reducing the need for dehumidifying procedures and water reposition, which results in lower water and energy consumption levels. Efficiency measures affecting the filter flow rates, preventing good water mixing by creating zones in which water cannot circulate, should be avoided because of users' health and safety [11,40].

Regular swimming pool filter backwashing is unavoidable due to the need to remove the contaminants accumulated during the filtration process and to condition the filter beds for continued efficient operation [41]. Nevertheless, this process generates large amounts of water, which may constitute 20% to 70% of the total facility wastewater volume discharged into the sewage contributing to water (and energy) use inefficiencies and environmental contamination [24]. Therefore, it is crucial to promote the sustainable

management of backwashed water. The preliminary results showed that this water could not be directly reused for irrigation because of its high concentration of TSS, free residual chlorine, and turbidity. However, the sedimentation process seemed to effectively reduce these parameters to permissible values, allowing the reuse of this water for irrigation purposes or for flushing cisterns, according to the parameters proposed in the available regulation [31–33]. Similar trends were found by [21–24]. Furthermore, these authors also verified that the sedimentation process was effective in removing nitrogen and phosphorus. Therefore, considering these results, it is plausible to assume that the management of filter washings is possible in this facility after the installation of a relatively simple system for their treatment, such as a settler or a settling tank. Furthermore, considering that the annual amount of water necessary for irrigating the lawn of the local football stadium, located in the surroundings of the swimming pool complex, is around 7200 m$^3$ (data provided by municipality), the backwashed water could be fully reused for this purpose. Nevertheless, the treated filter backwashing could also be used in the flushing cisterns or, after a broader treatment, reintroduced back into the swimming pool [41]. However, due to the lack of specific legislation for backwashing monitoring and quality control, other indicators such as nutrients, micropollutants, and other chlorine forms [41,42] should be determined in future approaches.

## 5. Conclusions

Under a climate change scenario, the improvement of water use efficiency is paramount. The present study obtained valuable information about the feasibility of promoting water efficiency in public swimming pools by implementing simple and relatively inexpensive, cost-effective measures, significantly reducing water consumption without jeopardizing users' comfort and health. Besides, the possibility of reusing the backwashed water increases water and energy efficiency even further. Furthermore, the methodology used in the present approach is easily replicated, allowing its application in any swimming pool complex. Nevertheless, future approaches are needed to: (1) determine the—water–energy nexus with more accuracy for this building to plan and manage the water and energy in an integrated manner, since there is an intrinsic link between both items; (2) to evaluate the feasibility of reusing the filter washings for irrigation and, eventually, for other purposes, in terms of the water–energy nexus, water quality, and investment turnover; and (3) develop educational measures to promote users' awareness of the importance of water and energy efficiency.

**Author Contributions:** Conceptualization, F.S., F.F., A.M.A.-G. and A.A.; methodology, F.S.; formal analysis, F.S., C.Z. and A.M.A.-G.; investigation, C.Z., M.J.A. and F.S.; resources, F.S.; data curation, F.S., C.Z. and A.M.A.-G.; writing—original draft preparation, C.Z., F.S. and A.M.A.-G.; writing—review and editing, A.M.A.-G., F.S. and A.A.; visualization A.M.A.-G., F.S. and A.A.; supervision, F.S., F.F. and A.A.; project administration, F.S. All authors have read and agreed to the published version of the manuscript.

**Funding:** This research received no external funding.

**Institutional Review Board Statement:** Not applicable.

**Informed Consent Statement:** Not applicable.

**Acknowledgments:** The authors are grateful to the Foundation for Science and Technology (FCT, Portugal) for financial support by national funds FCT/MCTES to CIMO (UIDB/00690/2020) and FibEnTech, (UIDB/00195/2020). Facilities provided by CMB were appreciated. Thanks are due to an anonymous reviewer for the comments and suggestions that helped to significantly improve the manuscript.

**Conflicts of Interest:** The authors declare no conflict of interest.

**Appendix A**

## Survey

This eight-question survey distributed to swimming pool complex users aimed to collate knowledge of the users' behavioral patterns concerning water use to calculate the total average flow of each device.

1. Gender: ☐ Male ☐ Female
2. Age:
3. Indicate the activities that you usually attend in the swimming pool complex and how often you participate in these activities per week:
    - ☐ Aqua Fitness for adults
    - ☐ Learning and swimming confidence for adults
    - ☐ Learning and swimming confidence for children and young people
    - ☐ Swimming confidence for children and young people
    - ☐ Infant swimming
    - ☐ Training
    - ☐ Other activity. Which?
4. How often do you wash your hands? ___________
5. How often do you use the shower? ___________

    5.1 Indicate the amount of time (in minutes) you spend in the shower: ___________

    Answer questions 6, 6.1 and 6.2 if you are male

6. How often do you use the following per day: Urinal: ___________ Toilet: ___________

    6.1 Indicate the number of flushes each time you use the urinal: _________
    6.2 Indicate the number of flushes each time you use the toilet: _________

    Answer questions 7 and 7.1 if you are female

7. How often do you use the toilet per day? ___________

    7.1 Indicate the number of flushes each time you use the toilet: _________

8. What measures do you think that should be implemented in the swimming pool complex to improve water efficiency?

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
