# Peer review of "Improving Water Efficiency in a Municipal Indoor Swimming-Pool Complex: A Case Study"

_applsci, doi:10.3390/app112210530_

Round 1

Reviewer 1 Report

Dear Authors, 

please check the text and make it easier to read and follow for everyone.

Overall, the presented results in Your study are interesting.

Author Response

Dear Reviewer,

The text was fully revised and improved.

The authors take this opportunity to express our thanks for the positive feedback and helpful comments for corrections or modifications.

Reviewer 2 Report

I summarize my comments as follows:

The article presents an interesting case study of a topic that is very understudied in the literature, but is of critical importance in the current context of water scarcity and climate change. Indeed the study focuses on the determination of water demand of a municipal swimming pool complex and propose water use efficiency measures for water water saving approach.

I want to thank the authors for this work and I encourage them for other studies in this field. 

Author Response

Dear Reviewer,

The text was fully revised and improved.

The Authors take this opportunity to express our thanks for the positive feedback and helpful comments for corrections or modifications.

Reviewer 3 Report

I like your introduction, but I suggest cutting parts of it, as you first start writing about swimming pools in line 47. You may add some references about swimming facilities being among the most energy-consuming building categories. The following references are proposed;

  1. Energy use and perceived health in indoor swimming pool facilities 1088/1757-899X/609/4/042051 (2019) and
  2. Energy Efficiency in Swimming Facilities. https://ntnuopen.ntnu.no/ntnu-xmlui/handle/ 11250/2366793 (2015)

It is also essential to mention how different cleaning measures and strategies for water supply affect the indoor air- and water quality in swimming facilities. Recent research has shown that pool management and disinfection methods may significantly affect the indoor air and water quality- thus also the health of the users in the poolroom. These parameters should never be compromised for water/energy saving measures; https://doi.org/10.1016/j.scitotenv.2020.138070 and DOI: 10.2166/wst.2019.291.

I believe you will increase your audience if you also focus on increase the water efficiency without compromising the water quality, as a sustainable solution should be social (clean and healthy water for everyone), profitable and environmental (reduced energy cost and energy use for water supply) .

How much fresh water is added per bather per day? Do such requirements exist in Portugal?

What is the water hydraulic retention time?

Line 151 – 163: In terms of air- and water quality, the concentration of combined chlorine is of importance. You measured free chlorine, turbidity, total suspended solids, pH-value, but did you not measure the concentration of combined chlorine? Did you not measure the water temperature?

Line 169- 171: “Although it is not possible to reduce water consumption in the pools and the filters backwashing process, it is feasible to promote more efficient use of water through reducing water consumption by adopting simple water-saving initiatives concerning personal use (showers, toilets, etc.)”. Are you sure there is nothing we can do to reduce the water consumption in the pools? What about reducing the water temperature, optimizing the relationship between relative humidity and water temperature, reduce the amount of aerosol generating activities (if such exist)?

Line 177: Please provide information about which devices you are referring to in the first sentence.

Line 213: I would remove the zeros from “5.00 L/min” as you also write “8 L/min”.

Line 251: Please provide information about which filters you are talking about in line 251.

Table 6: Please provide information about combined chlorine. It is great to propose strategies for improved water efficiency, but it should not compromise the water quality.

Round 2

Reviewer 3 Report

Thank you for considering all my comments. I see you have added total chlorine and combined chlorine to Table 1. I would prefer either total chlorine or combined chlorine also to be added to Table 5, but I understand from your comments that this was not measured. I also see you have added trihalomethanes (THM) to Table 1. I suggest you also add the method used to measure THM in  the water. 

Author Response

Dear Reviewer, 

Once more, thank you for the pertinent comments and suggestions. Indication of the methodology for trihalomethanes determination was included in the manuscript. English was fully revised.

Sincerely 

The Authors